# Trauma Experiences Are Common in Anorexia Nervosa and Related to Eating Disorder Pathology but Do Not Influence Weight-Gain during the Start of Treatment

**DOI:** 10.3390/jpm13050709

**Published:** 2023-04-23

**Authors:** Magnus Sjögren, Mia Beck Lichtenstein, Rene Klinkby Støving

**Affiliations:** 1Eating Disorder Research Unit, Psychiatric Center Ballerup, 2750 Ballerup, Denmark; 2Institute for Clinical Science, Sundsvall Regional Hospital, Umeå University, 851 86 Sundsvall, Sweden; 3Center for Eating Disorders, Odense University Hospital, 5000 Odense, Denmark; mlichtenstein@health.sdu.dk (M.B.L.); rene.k.stoving@gmail.com (R.K.S.); 4Research Unit for Medical Endocrinology, Odense University Hospital, 5230 Odense, Denmark; 5Mental Health Services in the Region of Southern Denmark, 5230 Odense, Denmark; 6Clinical Institute, University of Southern Denmark, 5000 Odense, Denmark

**Keywords:** anorexia nervosa, weight restoration treatment, traumatic experiences, eating disorder symptoms

## Abstract

Objective: The main characteristics of Anorexia Nervosa (AN) in adults are restriction of energy intake relative to requirements leading to significant weight loss, disturbed body image, and intense fear of becoming fat. Traumatic experiences (TE) have been reported as common, although less is known about the relationship with other symptoms in severe AN. We investigated the presence of TE, PTSD, and the relation between TE, eating disorder (ED) symptoms, and other symptoms in moderate to severe AN (*n* = 97) at admission to inpatient weight-restoration treatment. All patients were enrolled in the Prospective Longitudinal all-comer inclusion study on Eating Disorders (PROLED). Methods: TE were assessed using the Post-traumatic stress disorder checklist, Civilian version (PCL-C), and ED symptoms using the Eating Disorder Examination Questionnaire (EDE-Q); depressive symptoms were assessed using the Major Depression Inventory (MDI), and the presence of Post-traumatic Stress Disorder (PTSD) was diagnosed according to ICD-10 criteria. Results: The mean score on PCL-C was high (mean 44.6 SD 14.7), with 51% having a PCL-C score at or above 44 (*n* = 49, suggested cut-off for PTSD), although only one individual was clinically diagnosed with PTSD. There was a positive correlation between baseline scores of PCL-C and EDE-Q-global score (r = 0.43; *p* < 0.01) as well as of PCL-C and all EDE-Q subscores. None of the included patients were admitted for treatment of TE/PTSD during the first 8 weeks of treatment. Conclusions: In a group of patients with moderate to severe AN, TE were common, and scores were high, although only one had a diagnosis of PTSD. TE were related to ED symptoms at baseline, but this association diminished during the weight restoration treatment.

## 1. Introduction

Despite many years of research, the health threat of Anorexia Nervosa (AN) to young females remains high. The mortality rate is among the highest of any psychiatric disorder, surpassed only by opioid use disorder [1], with a risk nearly sixfold compared to a normal population [2,3], and for survivors, chronicity rates are high [4,5]. In adults, the disorders are described as multifactorial [6], presenting themselves with restriction in energy intake leading to significant weight loss, being distressed by one’s body weight or shape, and with an intense fear of becoming fat [6,7]. Accompanying features include reduced food intake, excessive exercise, and/or purging [8], which serve to reduce and maintain low body weight. The frequency of medical complications is high and often long-lasting [9]. The disorder also affects men, albeit to a lesser extent [10].

Both medical and psychological treatments are commonly used for AN, although no approved interventions for adults with AN exist. Enhanced cognitive behavioral therapy focal psychodynamic psychotherapy, the Maudsley model of AN treatment for adults, and specialist supportive clinical management are often recommended, while the evidence base is graded weak [6], with none having a superiority over the other [6]. For many patients with AN, weight restoration treatment is required, especially when weight is becoming alarmingly low, and in the short term, weight restoration has been found to improve prognosis [11]. However, comorbidities may influence treatment decisions, and a thorough investigation of concomitant morbidity is recommended by current treatment guidelines.

In different samples, the prevalence of PTSD in ED has been found to range from 1% to 52% [12,13,14,15,16], with the wide range being explained by differences in the PTSD criteria, the methods of assessment and recruitment, and whether cross-sectional or longitudinal/lifetime study designs were used. For example, the prevalence of PTSD in AN has been estimated to be between 10% [15] and 47% [13] in clinical samples. For comparison, in the National Women’s Study, a lifetime prevalence of PTSD was found in 11.8% of women with no eating disorder (ED) [12]. Moreover, in AN patients with a comorbidity of PTSD, this was more commonly found in the binge–purge subtype of AN than in the restricting subtype of AN [17]. Other studies have corroborated this finding, also describing TE to be more common in the binge–purge subtype of AN (AN-BP) than in the restricting subtype of AN (AN-R) [17,18,19].

The nature and frequency of the trauma have traditionally been difficult to characterize in ED since the definition and classification of traumatic events vary considerably across studies. Childhood sexual abuse [12,18,20,21], accidents, interpersonal loss or separation [22], and emotional abuse [23] have all been described in patients with ED [17]. Thereby, an assessment of the impact of TE on treatment will take this complexity into consideration. Multiple trauma may have influenced the current status, for example, by using an instrument to assess TE independently of the type or number of TEs, such as the post-traumatic stress disorder checklist (PCL).

TE may directly affect the development of ED. Several studies have found that both emotional and sexual abuse precede the development of AN [24,25,26,27], with proposed mechanisms involving emotional dysregulation and increased interoceptive awareness as consequences of TE [26,27]. Potentially, biological factors are involved as well, for example, increased stress as evidenced by an increase of the Hypothalamus–Pituitary–Adrenal axis activity in patients with childhood TE and AN [28,29]. Other studies have also found that TE and PTSD may exacerbate [30] and influence the severity of the ED [16,31]. Thus, TE is a key factor, both driving and influencing the expression of core symptoms in AN. In some studies, TE has also been found to negatively influence outcomes from treatment [32,33], although there are conflicting findings [34].

Only a few studies have investigated the presence of TE, PTSD, and their influence on ED symptoms in severe AN. Furthermore, the influence of TE on the outcome of inpatient treatment is even less known in severe AN. With data from a prospective study, the aim was to explore the presence of TE, PTSD, and the relation to ED symptoms at baseline and during weight restoration treatment in adult female and male patients with moderate to severe AN. Secondly, the aim was to explore how this may have influenced early treatment decisions.

## 2. Materials and Methods

### 2.1. Participants

Ninety-seven (*n* = 97) patients suffering from moderate to severe AN, ages ranging from 18 to 53 years, 93 women and 4 men, all meeting the ICD-10 and the DSM-5 criteria for AN [7,35] at the time of investigation [36], and a BMI < 17 (to ensure all were undergoing weight restoration treatment) were part of this analysis. Of these, seventy-seven (*n* = 77) had the restrictive type of AN (AN-R), and nineteen (*n* = 19) had the binge–purge subtype (AN-BP). Severity was based on the global level of functioning and symptoms [37]. All patients were partaking in the PROspective Longitudinal all-comer inclusion study on Eating Disorders (PROLED), which started in 2016, and is a clinical, longitudinal study planned to follow the patients annually over 10 years. All data in the current study was collected before May 2022. The study ran at the Psychiatric Center Ballerup (PCB) and was approved by the local ethics board (id: H-15012537; addendum 77106) and the data processing board. The general inclusion criteria in PROLED are as follows:-adult individuals (age 18–65 years);-admitted to the ED unit in Psychiatric Center Ballerup, Denmark;-a diagnosis of an ED;-a signed written informed consent.

Any patient that, at the time of screening, was undergoing forced care was excluded. The enrollment rate was 96% in 2016, 74% in 2017, 62% in 2018, and 68% in 2019. Data on referrals were collected from medical records.

### 2.2. Weight Restoration Treatment

All patients with a diagnosis of AN underwent a weight restoration program, which has been described previously [38,39,40]. In summary, meals were provided five times per day during monitoring by trained nurses to ensure proper renourishment. A dietician had individual weekly meetings with each patient to ensure a meal plan that would enable an approximate 1 kg weight increase per week up to an ideal body weight (IBW) of BMI 20 for women and BMI 21 for men. Weight gain was supported by restrictions in physical activity, monitored meals, and post-meal rest. Weekly measures of weight were conducted. All patients had undergone medical and psychiatric examinations, and any medical complications were addressed as they were identified. During these eight weeks, there was no formal psychotherapy provided, although individual meetings with psychologists and nurses for supportive reasons were offered to the patients. All patients received vitamins; however, no patient in this study underwent enteral feeding during this current course of treatment. In addition, trained physiotherapists offered a body relaxation program to all patients as part of the clinical inpatient program. An average stay at the ED unit was 10 weeks, independent of the reason for discharge. 

### 2.3. Clinical and Psychometric Measures

All patients underwent an initial complete diagnostic work-up, including a comprehensive diagnostic interview by a psychologist and medical and psychiatric examinations carried out by either a specialist psychiatrist or a General Practitioner with special training in EDs. In addition, the Eating Disorder Examination (diagnostic questions; EDE [41]) and routine clinical and laboratory assessments were done to maintain a high quality of diagnosing ED and comorbid disorders. All primary and, in case of any comorbid diagnoses, were validated using the ICD-10 checklist [42] and conducted by an independent (of the clinical diagnosing) physician.

The baseline and weekly assessments in PROLED have been described before [38,39,40]. Relevant to this analysis, assessments were done using the EDE-Q [43], the Post-traumatic stress disorder checklist Civilian (PCL-C) version [44,45], and the Major Depression Inventory (MDI) [46]. All were done at baseline, and EDE-Q was also done at discharge and MDI weekly until discharge. Only validated Danish versions of the instruments were used. 

The PCL-C is a self-report questionnaire consisting of 17 items, which are closely based on the DSM-IV criteria for PTSD. The PCL-C can be used in any population and asks questions about symptoms in relation to generic “stressful experiences”. This version simplifies assessment based on multiple traumas because symptom endorsements are not attributed to a specific event. The PCL has been found to have good temporal stability, high internal consistency, test–retest reliability, sensitivity, specificity, and convergent validity. In addition, the PCL-C scores are positively correlated with scores of the Mississippi PTSD Scale and The Minnesota Multiphasic Personality Inventory-2 (MMPI-2) Keane PTSD Scale. Setting the PCL-C cut-off score to 44 has been found to have a high sensitivity (0.94), specificity (0.86), and overall diagnostic efficiency (0.90) [44,45] for a diagnosis of PTSD.

The EDE-Q [43] is a self-report questionnaire that was developed from the investigator-based interview instrument, the Eating Disorder Examination (EDE). The EDE-Q is designed to measure the broad range of the specific psychopathology of EDs by measuring the present state of eating behavior and attitudes of subjects over the previous 28 days. The eating behavior and attitudes of subjects are assessed using a 7-point scale (from 0 to 6). The number of days a subject experienced a certain ED behavior is also scored. A high global score on EDE-Q indicates higher levels of problematic eating behaviors and attitudes. In the Danish version of the EDE-Q, subjects are requested to answer a total of 28 questions. Four sub-scales may be derived from its ratings: Dietary Restraint, Eating Concern, Weight Concern, and Shape Concern. A global score can be calculated from the averages of the four subscale scores.

The MDI [46] is a unidimensional self-rating instrument that covers the ICD-10 symptoms of depression for the past 14 days. Each item on the MDI gives a score between 0 and 5 on a Likert scale, and the scores are summed up with a score range between 0 to 50; the higher the total score, the more severe symptoms of depression.

### 2.4. Statistical Analysis

All analyses were conducted using the Statistical Package for Social Sciences (SPSS) Version 28. The distribution of the included data was assessed using the Shapiro–Wilks test, and any outliers and/or missing data were excluded by case. PCL-C scores were normally distributed, and thereby, parametric statistics were applied.

Clinical characteristics were described as means and standard deviation for age, duration, and BMI. Gender distribution was expressed in numbers and percentages. Correlations were done using the Pearson correlation test. A linear multiple regression analysis of baseline scores was done with EDE-Q global score as the dependent variable and age, duration, PCL-C score, and MDI score at baseline as independent variables. To investigate the effect of different factors on the change in EDE-Q global score over the course of treatment, a linear multiple regression analysis was done with change in EDE-Q global score as dependent, and age, duration, PCL-C score at baseline and change in MDI score, and change in weight during treatment, as independent variables.

One-way ANOVA was used to compare AN-R with AN-BP with regard to differences from treatment in a change in EDE-q, weight, and MDI, as well as in baseline scores of PCL-C. In addition, the AN subtype was also added to the linear regression analyses to explore the effect of the subtype on change in EDE-Q from treatment.

## 3. Results

### 3.1. Participants’ Baseline Clinical Characteristics, Trauma, and ED Psychopathology

For an overview of the descriptive data, see Table 1. In brief, 97 patients with AN, 4 males and 93 females, were included in the analysis. Their mean age was 27.0 years (SD = 10.0), a mean baseline BMI of 14.8 kg/m^2^ (SD = 1.3), and the mean duration of illness was 8.5 years (SD = 8.2). The BMI ranged from 11.9 to 17.0.

### 3.2. ED Psychopathology

As indicated by the mean global score of the EDE-Q in Table 1, the degree of ED symptoms was moderate to severe for most of the individuals. EDE-Q shape concern was especially high in this group of AN patients (mean 4.3, SD 1.5), followed by EDE-Q weight concern (mean 3.8, SD 1.6) and thereafter, EDE-Q restraint (mean 3.0, SD 1.8) and EDE-Q eating concerns (mean 2.9, SD 1.5).

### 3.3. Trauma

Only one patient had a diagnosis of PTSD, while the mean score of PCL-C was 44.6 (SD 14.7), which, according to a suggested cut-off of 44 for a diagnosis of PTSD, indicated a high level of TE severity in this AN sample with potentially several undiagnosed cases of PTSD. Indeed, 51% of all AN patients had a score on PCL-C of 44 or above. The PCL-C intrusion subscore mean was 11.6 (SD 5.8); the PCL-C avoidance subscore mean was 18.0 (SD 6.3), and the PCL-C hyperarousal subscore mean was 15.0 (4.8), suggesting that in this group of AN patients, the avoidance score was predominating. 

### 3.4. Change in ED and Other Psychopathology over Time of Treatment

The mean change in EDE-Q global scores from baseline to end of treatment was −0.53 (SD 1.3), and the mean change for MDI from baseline to endpoint was −6.6 (SD 10.5). 

### 3.5. Correlations at Baseline

Age and duration of illness were correlated (r = 0.74, *p* < 0.01), and PCL-C total score correlated with both EDE-Q global score (r = 0.43, *p* < 0.01) and MDI (r = 0.44, *p* < 0.01) at baseline. 

### 3.6. Linear Regression Analysis of Baseline Scores

A linear regression analysis with EDE-Q global score at baseline as the dependent variable and age, duration, PCL-C total score, and MDI, all at baseline, as independent variables, showed a significant effect of MDI and PCL-C (R^2^ = 3.6, SEE = 1.1), and an ANOVA gave an F = 24.5 (*p* < 0.001) with the coefficients for MDI (t = 4.7, *p* < 0.001) and PCL-C (t = 2.5, *p* < 0.05). 

### 3.7. Correlations between PCL-C at Baseline and Change in Psychopathology from Treatment

PCL-C total score at baseline did not correlate with either change in EDE-Q global score or a change in MDI score from baseline to endpoint. However, the change in EDE-Q global score was correlated with the change in the MDI score (r = 0.47, *p* < 0.01). 

### 3.8. Linear Regression Analysis of Change Scores from Treatment

A linear regression analysis with change in EDE-Q global score as a dependent variable and age, duration, PCL-C total score at baseline, and change in MDI in weight from treatment as independent variables showed a significant effect of MDI alone (R^2^ = 3.6, SEE = 1.3), and an ANOVA gave an F = 2.6 (*p* = 0.05), with the coefficients for MDI (t = 3.3, *p* < 0.01) and weigh change (t = −2.0, *p* = 0.058). 

### 3.9. Anorexia Nervosa Subgroup Analyses 

PCL-C score was found to be higher in AN-BP (*n* = 77, mean 51.4 SD 17.6) than in AN-R (*n* = 19, mean 43.0 SD 13.7; F = 5.0, *p* < 0.05). In AN-R, PCL-C was correlated with baseline EDE-Q global score (r = 0.31, *p* < 0.01) and MDI (r = 0.35, *p* < 0.01). In AN-BP, PCL-C was correlated to baseline EDE-Q global score (r = 0.70, *p* < 0.001) and MDI (r = 0.61, *p* < 0.01). There was no difference between AN-R and AN-BP from treatment in the change in EDE-Q global, weight, or MDI.

### 3.10. Referrals for Treatment of Comorbidities 

During the first 8 weeks of inpatient weight restoration treatment for patients with AN, none were referred to the treatment for concomitant morbidities. For patients with personality syndromes, DBT was offered, while none were referred for treatment of PTSD.

Figure 1: Plot of correlation between PCL-C score and EDE-Q total score in AN at baseline+.

## 4. Discussion

The frequency of PTSD in this study was similar to other AN studies [14], although some studies have found higher frequencies [15,17]. Using a cut-off of 44 in the PCL-C to diagnose PTSD in the current data set would vastly increase the frequency in the current study of PTSD to 51%, thereby casting some doubt as to the accuracy of the clinical diagnosis of PTSD in this set of AN patients. Potentially, other symptoms, such as anxiety, insomnia, phobias, social isolation, and more urgent medical needs, are obscuring the clinical picture and interfering with the work-up on differential diagnoses. The high scores of PCL-C in this study reveal that TE are common in AN, and although not always recognized as PTSD, they may warrant further attention. With the timepoint related to the development of ED described in some studies [23,31,33], interventions would optimally start close to the TE itself. However, in most adult patients with AN, the TE happened in the past, and both the timing, e.g., in relation to other interventions for AN, and the approach to the treatment of the TE, may not be trivial. In the current study, none were referred to treatment of TE. The clinical praxis during the first 8 weeks was to focus on weight increase and any medical complications or comorbidities and, to a lesser extent, psychiatric comorbidities. This may have explained the low referral rate. However, since the TE may influence treatment outcomes, at least in the long term, especially from psychological interventions, early recognition is warranted. Further research is needed on this topic. 

The results of this study suggest that TE are correlated to ED symptoms and depressive symptoms in AN. The correlation between TE and ED symptoms in this group of AN patients was positive and of moderate strength, implying that more TE correlated with more ED symptoms. We failed to find any previous studies in adults that described a correlation in AN patients undergoing weight restoration, although a similar correlation was found in some studies [21,47]. The mechanism behind this correlation cannot be elucidated from the current data set; however, it may be speculated that TE either drives, triggers, or both the ED symptoms, as several studies have suggested [24,25,26,27]. Alternatively, the presence of an ED may sensitize the individual to interpret experiences as traumatic. Longitudinal studies following individuals who are exposed to trauma and who later develop ED symptoms have found that childhood trauma is a risk factor for the development of an ED [23,31,33], which also relates to the severity of the ED [16,31], and even may exacerbate the disorder [30]. A potential biological factor that may either be a consequence or a maintaining factor, or both, is that patients with AN and TE present with a dysregulation of the HPA-axis activity [28,48]. 

In this set of moderate to severe patients with AN, and in spite of an improvement in both the ED symptoms and depressiveness over time, TE did not influence the outcome, i.e., change in ED symptoms from weight restoration treatment. To the best of our knowledge, no similar study in adult patients with AN undergoing weight restoration treatment has been presented. Although it would seem logical that TE would interfere with the ability to adhere to treatment in AN, e.g., due to higher distress and less ability to concentrate on therapy, it may be that this lack of influence on the change in ED symptoms relates to either the type of treatment provided, in this case being behavioral, focused on following commands from monitored food intake, the patients being in a structured inpatient environment, and that this itself provides relief over potential ambivalent thoughts that would be targeted in psychotherapy. Thereby, we cannot rule out that when other interventions such as psychotherapy are initiated, TE will start to interfere with treatment. 

While TE does not seem to impact weight restoration treatment, as evidenced by improvements in ED symptoms and depressiveness in AN, there was a clear correlation between depressiveness and ED symptoms, both at baseline and in relation to change in these two parameters from 8 weeks of treatment. This may imply that depressive symptoms and ED symptoms are more closely related than TE to ED symptoms, a finding also described by others, albeit using different study designs and interventions [47,49,50]. It may either be due to an effect of the weight restoration treatment itself or related factors, such as being in a structured environment, that make both ED symptoms and depressiveness improve. Alternatively, these symptoms are related to each other at a psychopathological level in AN. Since depression is a negative predictor of treatment outcome in AN, both in the short and long term [51]; depressiveness may reflect a part of the deeper psychopathology of AN and the paralleled change over time, such as in this study where both parameters improved during treatment, further supports this notion. 

One study found that the longer the exposure to TE, the longer the duration of the ED [52]. In the current study, no relation was found between the duration of disease and the score of TE, thereby casting some doubt on this notion. Differences in study design, populations assessed, and the type of trauma assessed may underlie these differences. 

Several other studies have found that TE are more common in AB-BP than in AN-R [17,53,54], and the current study corroborates this notion. Several studies have found a higher emotional dysregulation in AN-BP than in AN-R, which may reflect an underlying higher level of distress. Furthermore, treatments may affect AN subtypes differently [55], suggesting that patients with disparate subtypes of AN may benefit from receiving treatments that target triggering and maintaining factors in ways that are relevant to the specific AN subtype. Together with findings of an underlying disturbance in the HPA axis, potentially also divergent brain structural changes from AN-R [56], and a poorer outcome of AN-BP [57], this underscores the need for a better understanding of the factors driving the development of ED symptoms in AN and other ED, to find clues to more tailormade treatment.

The following limitations are of relevance. Firstly, the current dataset included adult patients with moderate to severe AN undergoing weight restoration treatment. Thereby, implications for children and adolescents, or milder degrees of AN, cannot be made. Secondly, as only four males were included; this dataset provides limited information on the larger group of males affected by AN. Furthermore, trauma was assessed using PCL-C, which has been validated and proven to capture different types of traumas to enable assessment of the composite effect of TE on other symptoms or outcomes from treatment. However, symptoms of TE may be complex, and utilizing a set of instruments and methods to elucidate and characterize the TE in more detail, including the potential effect of repeated trauma on the outcome, may have given additional information. Moreover, having a control group of age- and gender-matched healthy volunteers and performing a survey of the clinicians’ impressions of TE amongst the patients may also have yielded valuable information. In addition, it should be recognized that TE does not equal a diagnosis of PTSD.

## 5. Conclusions

The results of the current study with adult patients with moderate to severe AN undergoing weight restoration suggests a high frequency of TE at baseline, with almost 50% qualifying for a diagnosis of PTSD, according to the PCL-C scores. Furthermore, at baseline, there was a correlation between TE, ED symptoms, and depressive symptoms, i.e., the higher the TE, the more severe the ED symptoms. However, with time, the relationship between TE and ED symptoms diminished. Albeit TE did not influence the outcome in the short term, a recognition of TE at admission may influence treatment decisions in the longer term. 

## Figures and Tables

**Figure 1 jpm-13-00709-f001:**
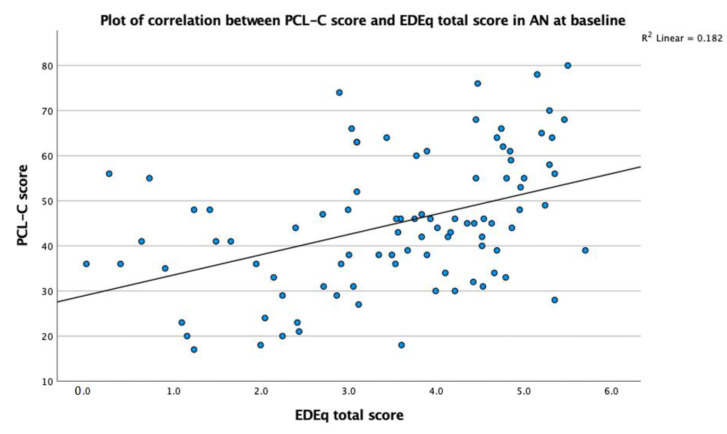
Plot of correlation between PCL-C score and EDE-Q total score in AN at baseline+.

**Table 1 jpm-13-00709-t001:** Participants’ (total *n* = 97, female = 93 and male = 4) baseline clinical characteristics.

Characteristics	Mean	SD
Age (years)	27.0	10.0
Duration of illness (years)	8.5	8.2
BMI baseline	14.8	1.3
EDE-Q-Restraint	3.0	1.8
EDE-Q-Eating concern	2.9	1.5
EDE-Q-Shape concern	4.3	1.5
EDE-Q-Weight concern	3.8	1.6
EDE-Q-Global score	3.5	1.4
MDI	31.5	9.1
PCL-C	44.6	14.7

## Data Availability

The datasets used in this study are currently not available since the prospective study is still ongoing.

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
