# Peer review of "Trauma Experiences Are Common in Anorexia Nervosa and Related to Eating Disorder Pathology but Do Not Influence Weight-Gain during the Start of Treatment"

_jpm, 2023, doi:10.3390/jpm13050709_

Round 1
Reviewer 1 Report (New Reviewer)
The manuscript is very interesting and represents an important insight into the care pathways of patients with ED.
However, I would ask the authors whether they also administered The Childhood Trauma Questionnaire (CTQ) to assess exposure to traumatic childhood experiences. If so, it would be very interesting to report it. PTSD is a clinically significant disorder, but often exposures to traumatic experiences in childhood (EDs often begin in adolescence and childhood traumas can have a causal relationship with EDs while not resulting in PTSD), do not always result in PTSD but impair mental health significantly.
A critical issue, from my point of view, is the choice to use the MDI for the assessment of depression. I would ask why this choice instead of conducting the assessment with the MADRS (Montgomery-Asberg Depression Rating Scale) or with the HAM-D (Hamilton Depression Rating Scale).
Author Response
We thank this peer for valuable feedback and comments. Below are our responses to the comments:
- We agree with this peer that The Childhood Trauma Questionnaire (CTQ) would have been an valuable addition to the questionnaires. However, this questionnaire, the CTQ, was not administered but only the PCL-C.
- We agree with this peer that administering an interview assessment such as HAM-D or MADRS would have been very valuable. However, this was the original ambition, but with time it became evident that only a few standardized and complete HAM-D assessments were available and not sufficient to be used for group analyses. However, we believe the strength of our design was the standardized administration of MDI, every Thursday after lunch while the patients were resting after meal intake. The MDI has also been validated towards HAM-D (P. Bech, N. Timmerby, K. Martiny, M. Lunde and S. Soendergaard. Psychometric evaluation of the Major Depression Inventory (MDI) as depression severity scale using the LEAD (Longitudinal Expert Assessment of All Data) as index of validity. BMC Psychiatry 2015 Vol. 15 Pages 190. Accession Number: 26242577 PMCID: PMC4526416 DOI: 10.1186/s12888-015-0529-3)
Reviewer 2 Report (New Reviewer)
This is an interesting and well-written manuscript reporting a clinically relevant study.
I have only minor comments:
- Regarding the affiliation, it is uncommon to write "former head of Eating Disorder research unit, Psychiatric Center Ballerup, Ballerup, Denmark". As the study was performed at the Psychiatric Center Ballerup, the authors should only write: "Eating Disorder Research Unit, Psychiatric Center Ballerup, Ballerup, Denmark."
- In the abstract, the authors might consider writing; "TE was related to ED 29 symptoms at baseline, but this association diminished during weight restoration treatment." The word "association" would make it easier to understand the sentence.
- Some limitations are mentioned in the discussion, but there are further limitations that should be listed:
1) The lack of an age- and sex-matched healthy control group, because we don't know whether people in the general population might have comparable traumatic experiences.
2) The authors conclude that there is a high level of traumatic experiences, but a lack of recognition of this by clinicians. However, they did not ask the clinicians about the traumatic experiences of their patients, they only examined the patients. Thus, the clinicians should have been asked as well.
3) Whether PTSD is diagnosed or whether traumatic experiences are recognized are two different questions. Not every traumatic experience justifies the diagnosis of PTSD. However, the study design does not allow to examine whether the traumatic experiences were recognized.
- I don't think the study justifies the conclusion that "trauma-focused therapy should be considered." The authors did not test whether trauma-focused therapy improves the outcomes of the eating disorder treatment. Thus, this conclusion is too far-reaching. From my point of view, it can only be concluded from the data provided that traumatic experiences might worsen eating disorder symptoms. Further thoughts about how to improve therapy based on the presented data should be clearly labelled as speculations not as conclusions.
Author Response
We thank this peer for valuable feedback and comments. Below are our responses to the comments:
- we have changed the affiliation
- this (the term "association") has been edited in accordance with this peer's suggestion
- we agree and have added these limitation (age- and gender matched healthy control; asking the clinicians; and that TE does not equal PTSD).
- we agree about trauma-focused therapy and have edited this sentence in the discussion and conclusion.
Reviewer 3 Report (New Reviewer)
I must confess that I read this manuscript with interest.
The introduction is very good. I would suggest the authors to add, information that eating disorders also affect eg in minorities.
10.1186/s12889-022-14943-7
10.1136/medhum-2020-011847
The methodology is described correct
The results are presented correctly
The discussion is written correctly and is supported by current research data.
The conclusions are supported by the results.
Congrats!
no comments
Author Response
We thank this peer for valuable feedback and comments. Below are our responses to the comments:
- we have added the the one reference that deals with anorexia nervosa.
This manuscript is a resubmission of an earlier submission. The following is a list of the peer review reports and author responses from that submission.
Round 1
Reviewer 1 Report
Thank you for the opportunity to review this manuscript. I read the manuscript with great interest and have provided some suggestions that I hope the authors will find useful when revising their manuscript.
Overall, the study is generally well written. However, the justification for the research is not clear. The author needs to undertake a much more rigorous literature review to justify their research question and aims, citing other relevant studies which have been undertaken in this area. The author also needs to provide greater justification for their analyses. Presently, the manuscript reads like there was data available from a large dataset, and the author is slicing up the data without much thought into the rationale or justification. This may not be the case, but the authors needs to provide the reader with greater confidence that there is a justification for doing this research.
Author Response
response:
- We thank this peer for valuable comments. The justification has been specified in the abstract and the introduction. In addition, supporting evidence from the literature, has also been added.
- With regard to the literature, am updated search in PUBMED for publications on 1) "trauma" and 2) "Anorexia Nervosa" and 3) "weight", which yielded 95 publications. Out of these, 61 publications concerned adults. In addition, we searched using the following search terms in PUBMED 1) "abuse" and 2) "Anorexia Nervosa" and 3) "weight", and 4) "adult" which yielded 167 publications. Reading through all abstracts yielded no publication that described the influence or impact of trauma on eating disorder symptoms or weight change in adults with AN. There were a few publications that described the impact of traumatic events on the development of an ED, and the relation between childhood trauma/abuse and outcome from psychotherapy, and these have been added to the introduction.
Reviewer 2 Report
The purpose of this study was to examine the relation between ED symptoms, traumatic experiences, PTSD, and the effect of trauma on weight restoration in patients with moderate to severe AN. Trauma was found to be common but did not have an impact on weight restoration. This paper has the potential to be an interesting addition to the literature, but the authors tend to overstate their findings, and do not do an adequate job of reviewing the vast literature on trauma and EDs. I have several other concerns/suggestions:
Abstract: In the first sentence, the main characteristics of AN are not quite correct. While AN could involve weight loss, it could also involve failure to gain weight when appropriate. Excessive exercise and/or compensatory behavior could be involved but are not part of the diagnostic criteria for AN. This sentence should be rewritten in line with the DSM-5 criteria for AN.
Introduction, lines 35-36: AN now has the second highest rate of mortality, behind opioid use disorders (Chesney et al., 2014).
Introduction, lines 37-40: As in the abstract, these sentences should be adjusted to more accurately reflect DSM-5 criteria for AN.
Introduction, lines 42-43: There are in fact approved interventions for AN. Family-based treatment (FBT) for adolescents has a good evidence base and is recommended by several national guidelines, such as NICE. If this study is focused on adults, for whom there is not a first-line treatment, then this should be made clear in the manuscript.
Introduction, lines 62-64: The authors mention the importance of examining the impact of multiple traumas. Why did they decide not to examine this in the current study?
Introduction, lines 69-71: This sentence is unclear and would benefit from some explanation. What does “the embodiment of traumatic experiences” mean exactly? And how is this related to body image?
Introduction, line 76: Can the authors be more specific in their hypothesis, rather than using the word “frequent”? What counts as frequent?
Introduction, hypotheses: Why did the authors only look at change in ED symptoms and not change in weight as an outcome? Also, the hypothesis that PCL-C scores would be related to body image needs a better rationale, laid out in the introduction. Also, given that trauma has been found to be higher among patients with AN-BP than AN-R, why were there no differences expected or examined between diagnoses?
Weight restoration treatment, line 103: Can the authors provide a bit more information about the program, rather than just including references?
Clinical and psychometric measures, line 112: It seems that the EDE was conducted at the initial assessment. Why was it not repeated at discharge?
Clinical and psychometric measures, lines 137-138: This sentence is not quite correct. It is true that “points are measured in terms of the number of days those subjects experience a certain ED behavior”, but the behavioral items are not included in the EDE-Q subscales, so the second part of the sentence, “so that high EDE-Q scores indicate high levels of ED pathology” does not follow from the first part.
Clinical and psychometric measures, lines 141-142: This sentence should be reworded. Perhaps, “A global score can be calculated from the averages of the four subscale scores.”
Clinical and psychometric measures, line 145: What does “theoretical” mean in this context?
Table 1: The BMI range is confusing – it is 11.9 to 17.0? This needs to be changed to make it more clear.
Results, ED psychopathology, line 167: I’m not sure I would describe baseline EDE-Q scores as “moderate-severe to severe”. Only one subscale was above the often-used cutoff of 4.
Results, Change in psychopathology over time of treatment: Does this “mean change” in EDE-Q scores refer to the Global Score?
Results, Regression analysis: This is a confusing paragraph. Please explain more clearly what exactly the analyses were trying to determine. Were age, duration of illness, PCL-C score and MDI score the IVs?
Discussion, line 214: I might use a different word than “strong” to describe the correlations, unless the authors have a citation indicating that a correlation of 0.4 can be considered “strong”.
Discussion, lines 216-217: I’m not sure this statement is accurate. There are many studies on EDs and trauma. In general, I’m not sure that the authors have adequately reviewed the many studies on this topic.
Discussion, lines 234-235: Can the authors explain more clearly how the symptoms of AN may mimic PTSD?
Discussion, line 241: What does “trivial” mean in this context?
Discussion, line 245: The authors state that “traumatic experiences did not influence the results of weight restoration treatment”. This is not accurate. Trauma did not influence ED symptoms, but the authors did not report on whether it impacted weight restoration.
Discussion, lines 245-247: The authors seem to say that frequent assessments during treatment was a strength of the study, but unless I missed it, these were not used in the statistical analyses.
Discussion, lines 267-269: These sentences need to be reworded. First, this one study by itself does not refute previous studies. Second, this study did not assess the impact of multiple traumas.
Discussion, lines 275-276: This sentence is unclear. Are the authors suggesting that patients with AN-BP might be less compliant with treatment?
Discussion, line 278: What does “embodied” mean here? This needs to be elaborated upon.
Discussion, lines 279-282: Once again, this one finding does not refute any hypotheses. Furthermore, the current study did not assess body image. The EDE-Q subscales of shape and weight concern may overlap with body image, but they are not interchangeable.
Discussion, lines 288-290: The PCL-C does not limit participants to answer questions in response to only one traumatic event, but it also is not able to assess the impact of multiple types of trauma.
Conclusion, line 299: It is not accurate to say, “traumatic experiences seem to have a weaker effect on outcome” if no relation was found.
Conclusion, last line: It is also not accurate to say that “efforts to reduce the depressiveness are more impactful”. That was not a finding of this study.
There are several grammatical errors that should be corrected prior to publication.
Author Response
The purpose of this study was to examine the relation between ED symptoms, traumatic experiences, PTSD, and the effect of trauma on weight restoration in patients with moderate to severe AN. Trauma was found to be common but did not have an impact on weight restoration. This paper has the potential to be an interesting addition to the literature, but the authors tend to overstate their findings, and do not do an adequate job of reviewing the vast literature on trauma and EDs. I have several other concerns/suggestions:
- With regard to the literature, am updated search in PUBMED for publications on 1) "trauma" and 2) "Anorexia Nervosa" and 3) "weight", which yielded 95 publications. Out of these, 61 publications concerned adults. ..., AND, in addition, we searched using the following ... updated search in PUBMED for publications on 1) "abuse" and 2) "Anorexia Nervosa" and 3) "weight", and 4) "adult" which yielded 167 publications. Reading through all abstracts yielded no publication that described the influence or impact of trauma on eating disorder symptoms or weight change in adults with AN. There were a few publications that described the impact of traumatic events on the development of an ED, and the relation between childhood trauma/abuse and outcome from psychotherapy, and these have been added to the introduction. If there are additional publications that this search, and previous, have missed, we are eager to learn about these.
- Abstract: In the first sentence, the main characteristics of AN are not quite correct. While AN could involve weight loss, it could also involve failure to gain weight when appropriate. Excessive exercise and/or compensatory behavior could be involved but are not part of the diagnostic criteria for AN. This sentence should be rewritten in line with the DSM-5 criteria for AN.
- This study was done in adults and thereby, only weight loss is relevant. We have specified this in the abstract.
- In addition, the study was done in EU and while many clinicians and scientists use the DSM-5, this study was following patient in clinical practice in Denmark where the the ICD-10 is used. This is specified in the methods and in the abstract. We have however updated the text in the abstract to more closely follow DSM criteria that are align with ICD10..
- Introduction, lines 35-36: AN now has the second highest rate of mortality, behind opioid use disorders (Chesney et al., 2014).
- This has been corrected.
- Introduction, lines 37-40: As in the abstract, these sentences should be adjusted to more accurately reflect DSM-5 criteria for AN.
- As mentioned above, this study was done in Denmark and following the ICD-10.
- Introduction, lines 42-43: There are in fact approved interventions for AN. Family-based treatment (FBT) for adolescents has a good evidence base and is recommended by several national guidelines, such as NICE. If this study is focused on adults, for whom there is not a first-line treatment, then this should be made clear in the manuscript.
- Indeed, this was already mentioned in the original submitted manuscript that this study only included adults. However, in order to make it even more clear, we have added this in the abstract. That is also the reason we are not mentioning FBT.
- Introduction, lines 62-64: The authors mention the importance of examining the impact of multiple traumas. Why did they decide not to examine this in the current study?
- We have clarified this in the introduction and the methods. The aim of this study was not to investigate the effect of the number of traumatic events, while we acknowledge that patients may have suffered one or several TE. Therefore, we used an instrument, PCL-C, that measures symptoms of PTSD independantly of whether patients had suffered one or more TEs.
- Introduction, lines 69-71: This sentence is unclear and would benefit from some explanation. What does “the embodiment of traumatic experiences” mean exactly? And how is this related to body image?
- We agree that this is unclear and have removed this exploratory idea.
- Introduction, line 76: Can the authors be more specific in their hypothesis, rather than using the word “frequent”? What counts as frequent?
- We have rewritten this using referring to the scores of the AN group.
- Introduction, hypotheses: Why did the authors only look at change in ED symptoms and not change in weight as an outcome? Also, the hypothesis that PCL-C scores would be related to body image needs a better rationale, laid out in the introduction. Also, given that trauma has been found to be higher among patients with AN-BP than AN-R, why were there no differences expected or examined between diagnoses?
- The aim originally was to study the relation between traumatic experiences and change in ED symptoms from treatment. We agree that it is of value to include weight change and have added an analysis of this. See results and discussion.
- We have removed the exploratory hypothesis that traumatic experiences would affect body image (see above).
- We did also want to know the difference between AN-R and AN-BP, why this was included in the original version of the manuscript (in the introduction) and an analysis (results and discussion). This has now been further stressed in this edited version of the manuscript in the introduction. In addition, in the subgroup analysis on AN subtype (ANR vs ANBP), we have also included we have also added the impact of subgroup of AN on ED symptoms from weight restoration treatment.
- Weight restoration treatment, line 103: Can the authors provide a bit more information about the program, rather than just including references?
- It is standard to refer to previous publications in order not to replicate the same information about methods in every publication stemming from e.g. a prospective study. However, we have now added this information also to this publication.
- Clinical and psychometric measures, line 112: It seems that the EDE was conducted at the initial assessment. Why was it not repeated at discharge?
- The reason for this was lack of resources. In addition, only the EDE diagnostic items were used and thereby, although diagnostic changes may occur during treatment, this was not the purpose of neither PROLED nor the current study. Instead, we had many self-questionnaires that patients did both at baseline, during treatment and at endpoint, which has provided valuable information.
- Clinical and psychometric measures, lines 137-138: This sentence is not quite correct. It is true that “points are measured in terms of the number of days those subjects experience a certain ED behavior”, but the behavioral items are not included in the EDE-Q subscales, so the second part of the sentence, “so that high EDE-Q scores indicate high levels of ED pathology” does not follow from the first part.
- we have corrected accordingly.
- Clinical and psychometric measures, lines 141-142: This sentence should be reworded. Perhaps, “A global score can be calculated from the averages of the four subscale scores.”
- We have corrected accordingly.
- Clinical and psychometric measures, line 145: What does “theoretical” mean in this context?
- In this context, the meaning of "theoretical" was "possible". We have deleted the word since it does not help explain.
- Table 1: The BMI range is confusing – it is 11.9 to 17.0? This needs to be changed to make it more clear.
- we agree that this was confusing and have deleted this row and moved the information to the results section (text).
- Results, ED psychopathology, line 167: I’m not sure I would describe baseline EDE-Q scores as “moderate-severe to severe”. Only one subscale was above the often-used cutoff of 4.
- We agree and have corrected this.
- Results, Change in psychopathology over time of treatment: Does this “mean change” in EDE-Q scores refer to the Global Score?
- yes. we have specified this.
- Results, Regression analysis: This is a confusing paragraph. Please explain more clearly what exactly the analyses were trying to determine. Were age, duration of illness, PCL-C score and MDI score the IVs?
- we agree this was too brief and have added a paragraph in the "statistics" section and in the "results" to better explain hat was done.
- Discussion, line 214: I might use a different word than “strong” to describe the correlations, unless the authors have a citation indicating that a correlation of 0.4 can be considered “strong”.
- we agree and have changed this to "moderate"
- Discussion, lines 216-217: I’m not sure this statement is accurate. There are many studies on EDs and trauma. In general, I’m not sure that the authors have adequately reviewed the many studies on this topic.
- see the response above. We have specified that this relates to adults undergoing weight restoration, although even with the new literature search included (described above) we did not find any additional publications describing this.
- Discussion, lines 234-235: Can the authors explain more clearly how the symptoms of AN may mimic PTSD?
- We have rewritten this section and included some symptoms.
- Discussion, line 241: What does “trivial” mean in this context?
- "straight forward".
- Discussion, line 245: The authors state that “traumatic experiences did not influence the results of weight restoration treatment”. This is not accurate. Trauma did not influence ED symptoms, but the authors did not report on whether it impacted weight restoration.
- We have included weight in updated regression analyses. Please see results. However, our aim was to study the influence on ED symptoms, as a primary outcome, and not weight gain.
- Discussion, lines 245-247: The authors seem to say that frequent assessments during treatment was a strength of the study, but unless I missed it, these were not used in the statistical analyses.
- That is correct. The sentence has been corrected.
- Discussion, lines 267-269: These sentences need to be reworded. First, this one study by itself does not refute previous studies. Second, this study did not assess the impact of multiple traumas.
- We agree and have deleted this whole section, since it was an exploratory hypothesis and not relevant for the main aims of this study.
- Discussion, lines 275-276: This sentence is unclear. Are the authors suggesting that patients with AN-BP might be less compliant with treatment?
- We agree that this was taken out of context. We have edited this section.
- Discussion, line 278: What does “embodied” mean here? This needs to be elaborated upon.
- we have deleted this section.
- Discussion, lines 279-282: Once again, this one finding does not refute any hypotheses. Furthermore, the current study did not assess body image. The EDE-Q subscales of shape and weight concern may overlap with body image, but they are not interchangeable.
- We agree and have removed this.
- Discussion, lines 288-290: The PCL-C does not limit participants to answer questions in response to only one traumatic event, but it also is not able to assess the impact of multiple types of trauma.
- We agree and have added this information.
- Conclusion, line 299: It is not accurate to say, “traumatic experiences seem to have a weaker effect on outcome” if no relation was found.
- we agree and have changed this sentence
- Conclusion, last line: It is also not accurate to say that “efforts to reduce the depressiveness are more impactful”. That was not a finding of this study.
- we agree. the sentence has been removed.
- There are several grammatical errors that should be corrected prior to publication.
- The language has been updated.
Round 2
Reviewer 1 Report
My feedback has not been adequately addressed and I have concerns about how this study was carried out.
Reviewer 2 Report
Thank you for the revisions. The manuscript is much improved. I have just a couple of remaining comments:
In the abstract, the authors state that PTSD is uncommon, but then report rates of 11.8% in a non-ED sample in the introduction. These statements seem inconsistent with each other.
Results: What was the average BMI at end of treatment?
Discussion, lines 283-285: Please provide a reference here. In addition, the word “trivial” should be replaced. In their response, the authors say that “trivial” in this context means “straightforward”. “Straightforward” may be a better word to use here.
In addition, a few grammatical errors remain.